# GAETS: A GRAPH AUTOENCODER TIME SERIES APPROACH TOWARDS BATTERY PARAMETER ESTIMATION

## ABSTRACT

Lithium-ion batteries are powering the ongoing transportation electrification revolution. Lithium-ion batteries possess higher energy density and favourable electrochemical properties which make it a preferable energy source for electric vehicles. Precise estimation of battery parameters (Charge capacity, voltage etc) is vital to estimate the available range in an electric vehicle. Graph-based estimation techniques enable us to understand the variable dependencies underpinning them to improve estimates. In this paper we employ Graph Neural Networks for battery parameter estimation, we introduce a unique graph autoencoder time series estimation approach. Variables in battery measurements are known to have an underlying relationship with each other in a certain causal structure. Therefore, we include ideas from the field of causal structure learning as a regularisation to our learned adjacency matrix technique. We use graph autoencoder based on a non-linear version of NOTEARS Zheng et al. (2018) as this allowed us to perform gradient-descent in learning the structure (instead of treating it as a combinatorial optimisation problem). The proposed architecture outperforms the state-of-the-art Graph Time Series (GTS) Shang et al. (2021a) architecture for battery parameter estimation. We call our method GAETS (Graph AutoEncoder Time Series).

## 1 INTRODUCTION AND LITERATURE REVIEW

Energy dense Lithium-ion batteries (LiB) are driving the current electrification revolution in the transportation sector Balasingam & Pattipati (2021). Rapid advancements in battery material science have enabled inexpensive and long-running LiBs. However, capacity degradation leading to poor performance are some of the drawbacks of these batteries Zhou et al. (2021). Development of accurate State-of-Charge estimation algorithms have garnered attention over the past decade as these metrics translate to available range in an Electric Vehicle Herle et al. (2020).

Battery degradation or capacity fade occurs due to repetitive charge/discharge cycles. Decreased battery capacity over its usage is an indicator of poor health of the battery Mohtat et al. (2021). Present battery models available in literature suffer from reliable battery capacity fade predictions Hinz (2019). This uncertainty in prediction is chiefly due to the non-linear behaviour of battery discharge characteristics with increased battery age.

Battery Ageing has also been a topic of considerable interest in the energy storage domain Ning & Popov (2004); Ning et al. (2006); Peterson et al. (2010); Barré et al. (2013). Empirical battery models have been used for Remaining Useful Life prediction of batteries Saha & Goebel (2009); Daigle & Kulkarni (2016); Nuhic et al. (2018); Andre et al. (2013). Simple estimators from Kalman Filter Andre et al. (2013); Bole et al. (2014) to Reinforcement Learning techniques Unagar et al. (2020) have been employed for battery parameter estimation.

All the current battery parameter estimation techniques available in literature can be broadly classified as follows:

1. Physics-based modeling which employs Partial Differential Equations (PDEs) to compute equivalent battery parameters Wang et al. (2011)

2. Phenomenological battery modeling, achieved by focused computation of battery capacity changes corresponding to specific charge/discharge cycles Goebel et al. (2008)

3. Data-driven estimation models, which provide meaningful insights from historical battery data and do not involve computation of complex electrochemical models Severson et al. (2019)

In the present work we use a Graph-based data-driven technique to gain meaningful insights from battery datasets to improve estimate accuracy. The paper is organised as follows, a brief review of graph based forecasting techniques is covered in Section 2, the architecture of the proposed Graph-based estimation technique is presented in Section 3, details about experimentation are covered in Section 4 followed by conclusion in Section 5.

## 2 GRAPH-BASED TIME SERIES ESTIMATION: RELATED WORK

Battery datasets typically consist of time series Voltage, Current, Temperature and State-of-Charge values as shown in Fig. 1. Statistical Time series based estimation techniques have been popular Bhatnagar et al. (2021) for typical time series datasets.

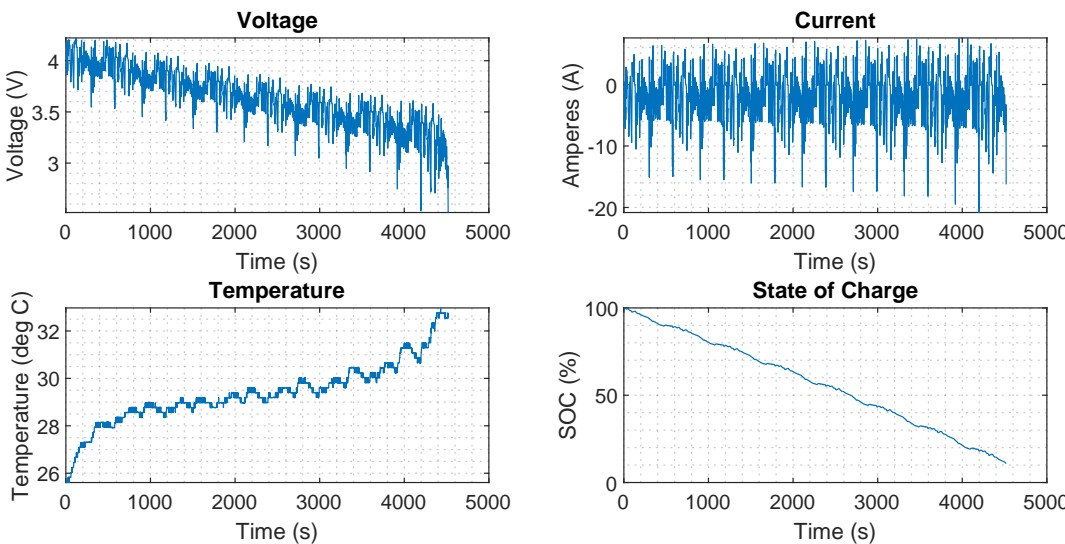

Figure 1: Typical battery discharge characteristics of 18650 cells Herle et al. (2020)

In Multivariate time series scenarios variable dependencies have been investigated to improve estimations Yu et al. (2018). Traffic datasets are one of the prime examples wherein variation in speed and traffic density insights are useful to accurately predict vehicular traffic in other nodes of the network. There are numerous Graph Neural Network (GNN) based techniques which exploit the graph structure of datasets to improve prediction accuracy Seo et al. (2016), Li et al. (2018) and Zhao et al. (2020).

All Graph-based forecasting and estimation approaches available in literature have been validated using traffic datasets. A more recent variant of Graph Neural Network dubbed GTS (Graph for Time Series) has been successfully employed to learn specific graph structures to improve prediction accuracy Shang et al. (2021b). In this paper we build upon the GTS architecture specifically for battery datasets. Following section elucidates the proposed architecture in greater detail.

## 3 GAETS ARCHITECTURE

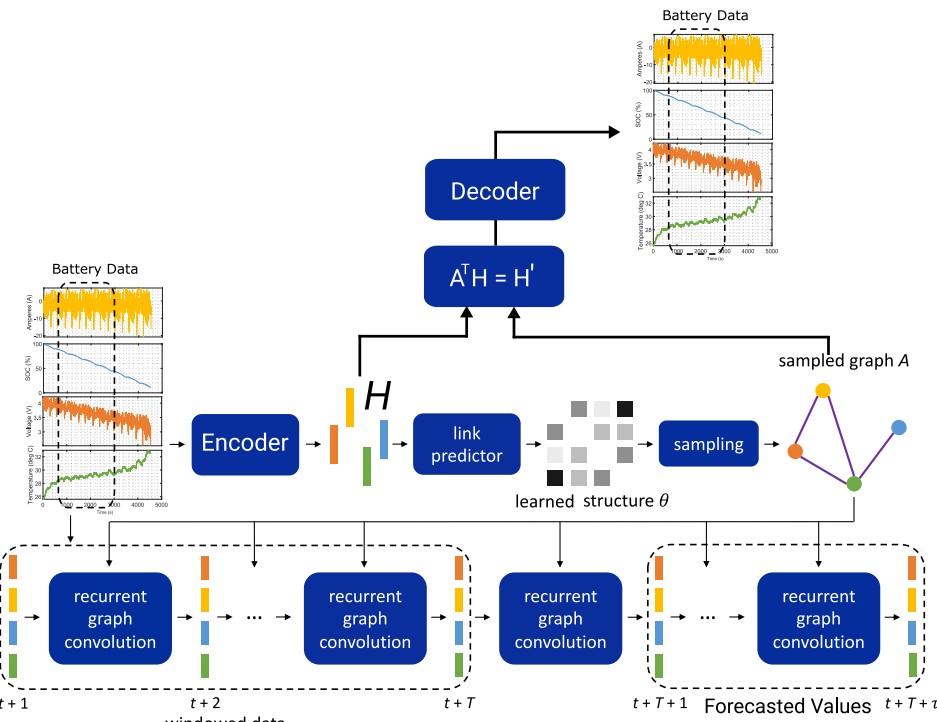

Figure 2: Graph AutoEncoder Time Series (GAETS) architecture

## 3.1 THE METHOD

The architecture of GAETS is as depicted in 2. The objective of the method is to forecast $\hat{X}_{t+T+1:t+T+\tau}$, given the observations $X_{t+1:t+T}$, where $X$ denotes training data and consists of multiple charge/discharge cycles of battery measurements. The model is denoted as $\hat{X}_{t+T+1:t+T+\tau} = f(A, w, X_{t+1:t+T})$, where $A$ is the graph structure and is parameterized by $w$, $T$ is the input window duration, our goal is to predict future data from time $t + T + 1$ to $t + T + \tau$, here $\tau$ is the output prediction window length. The graph structure represented by adjacency matrix $A \in \{0,1\}^{n \times n}$ is to be parameterized as it is exclusively not available to the forecasting model. GTS Shang et al. (2021b) parameterizes the graph structure by considering the adjacency matrix $A$ to be random variable on matrix Bernoulli distribution and is learned as part of the task. We use Gumbel reparameterization trick as it is used in Shang et al. (2021b) because the gradients fail to flow through $A$ in Bernoulli sampling.

### 3.1.1 ENCODER-DECODER

The encoder-decoder architecture is used in order to reconstruct the original time series data and to learn the intermediate represetations. The model assumes an inherent nonlinear Structural Equation Model (SEM) that connects the node features to the adjacency matrix. The SEM was first proposed in NOTEARS Zheng et al. (2018).

$$X^{(j)} = f(X^{(j)}, A) = g_2(A^T g_1(X^{(j)}))$$ (1)

where $g_1$ and $g_2$ are neural networks, encoder and decoder respectively while $j$ represents the node id. This can be seen as a graph autoencoder. The encoder extracts the features from the original time series data, by applying a fully connected layer on the convolved original features.

$$h^i = FC\left(vec\left(Conv\left(X^i\right)\right)\right)$$

where $h^i$ is the features extracted in the lower dimensional space. $X^i$ is the $i$th series of features. The matrix $H$ of representations $h^i$s generated by the encoders are used in the graph autoencoder as well as in the link predictor. In this scenario the vector representations are fed as input to the link predictor which spits out a scalar. The output scalar decides the existance of a link. These are represented as $h_i$ and $h_j$, the vectors are concatenated and passed through two fully connected layers to finally get the scalar.

The extracted features along with the sampled graph adjacency matrix is passed through the decoder. We make use of an MLP as the decoder. The loss incurred by the autoencoder also adds up to the final loss to be optimized.

### 3.1.2 RECURRENT NETWORKS

In order to estimate the time series predictions based on graphs, we make use of diffusion convolutional GRU as in DCRNN Li et al. (2018) and GTS Shang et al. (2021b).

$$R_{t'} = \sigma\left(W_{R\star A}\left[X_{t'} \parallel H_{t'-1}\right] + b_R\right), \quad C_{t'} = \tanh\left(W_{C\star A}\left[X_{t'} \parallel (R_{t'} \odot H_{t'-1})\right] + b_C\right),$$
$$U_{t'} = \sigma\left(W_{U\star A}\left[X_{t'} \parallel H_{t'-1}\right] + b_U\right), \quad H = U_{t'} \odot H_{t'-1} + (1 - U_{t'}) \odot C_{t'},$$

where

$$W_{Q\star A}Y = \sum_{k=0}^{K}\left(\theta_{k,1}^{Q}\left(D_O^{-1}A\right)^k + \theta_{k,2}^{Q}\left(D_I^{-1}A\right)^k\right)Y,$$

where all $W$s are corresponding weight matrices, $D_I$ and $D_O$ are in-degree and out-degree matrix respectively. $\theta_{k,1}^{Q}$ and $\theta_{k,2}$ and $b_Q$ are the model parameters and $K$ is the hyperparameter corresponding to the degree of diffusion.

### 3.2 TRAINING

The time series forecasting loss is the base loss which is the loss between the values forecasted by the models and the ground truth. Hence the first objective is given in 2.

$$\min l_{base}^{t}(\hat{X}_{t+T+1:t+T+\tau}) = \frac{1}{\tau}\sum_{t'=t+T+1}^{t+T+\tau}\left\|\hat{X}_t' - X_{t'}\right\| \tag{2}$$

The autoencoder compares the reconstructed data against the input data. The autoencoder loss, the second objective is given in 3.

$$\min l_{autoencoder} = \frac{1}{2n}\sum_{j=1}^{n}\left\|X^{(j)} - g_2(A^T g_1(X^{(j)}))\right\| \tag{3}$$

where, $n$ is the number of samples, and $\min l_{autoencoder}$ acts as a regularizer to the original GTS (Shang et al. (2021a)) loss. The original GTS loss is given in 2.

The final loss comprises of the loss incurred by the autoencoder and the base loss. It is given in 4.

$$loss = l_{base}^{t}(\hat{X}_{t+T+1:t+T+\tau}) + l_{autoencoder} \tag{4}$$

Finally, we arrive at the expression 5 that we need to minimize.

$$loss = \frac{1}{\tau}\sum_{t'=t+T+1}^{t+T+\tau}\left\|\hat{X}_t' - X_{t'}\right\| + \frac{1}{2n}\sum_{j=1}^{n}\left\|X^{(j)} - g_2(A^T g_1(X^{(j)}))\right\| \tag{5}$$

## 4 EXPERIMENTS

To validate the GAETS architecture we used Lithium-ion Battery Data (2018). All the batteries listed in the dataset had multiple charge/discharge cycles. We trained the model on two sets of measurement values logged from two separate channels. The data consists of 6 measurements namely: Voltage, Current, Charge Capacity, Discharge Capacity, Charge Energy and Discharge Energy. Each measurement is treated as a node and the task is to learn the structure that represents relationship between variables as well as forecasting the battery parameter of interest. We fix the input horizon to be 80 and test the data on three different forecasting window horizons of 40, 80 and 120.

In terms of tensor size, the training data is made of $(X_{train}, Y_{train})$ where $X_{train}$ is 1497 signals with a dimension of 6 (for each measurement) and horizon as 80, while $Y_{train}$ is 1497 signals with a dimension of 6 and 40/80/120 horizon values. Validation is done on 213 signals. Meanwhile, testing data is made of $(X_{test}, Y_{test})$ where $X_{test}$ is 1497 signals with a dimension of 6 (for each measurement) and 80 window ticks, while $Y_{test}$ is 1497 signals with a dimension of 6 and 40/80/120 horizon values. These datasets are obtained by continuous moving window average across the original long time series data.

We follow the code implementation in GTS Shang et al. (2021a), using the same setting for the graph neural network architecture and optimiser. We train each model for 200 epochs, obtain the best model for validation dataset and test it. Each model is trained from 5 different random initialisations. The performance result and confidence interval is shown in the chart. Performance is measured through MAE, RMSE and MAPE values as shown in Fig. 6 following GTS Shang et al. (2021a) paper. Visual comparison of Battery Voltage, Current and Charge Energy is presented in Fig. 3,4 and 5.

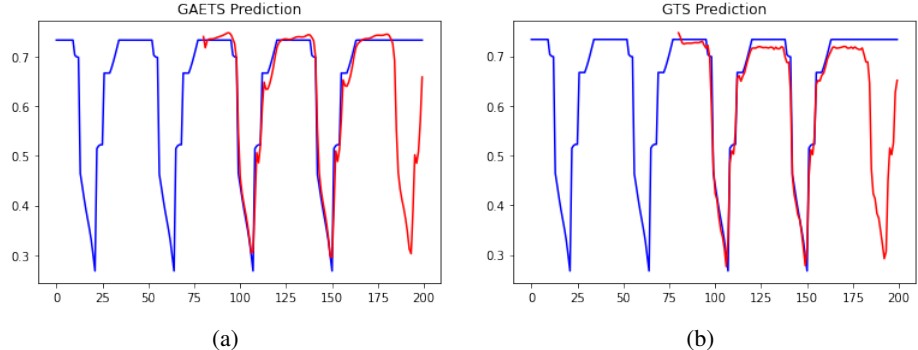

(a)                                                         (b)

Figure 3: Battery Voltage predictions using GAETS (this work) and GTS Shang et al. (2021a)

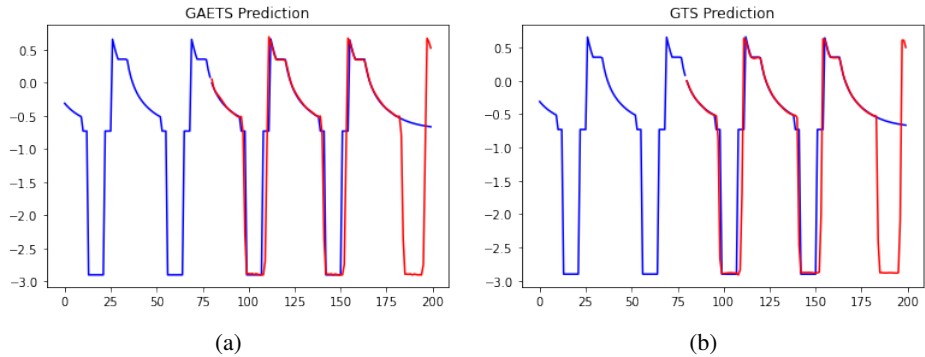

(a)                                                         (b)

Figure 4: Battery Current predictions using GAETS (this work) and GTS Shang et al. (2021a)

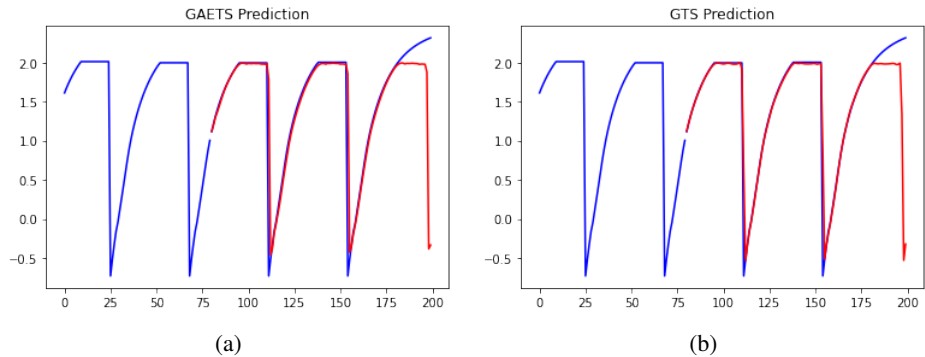

Figure 5: Battery Energy predictions using GAETS (this work) and GTS Shang et al. (2021a)

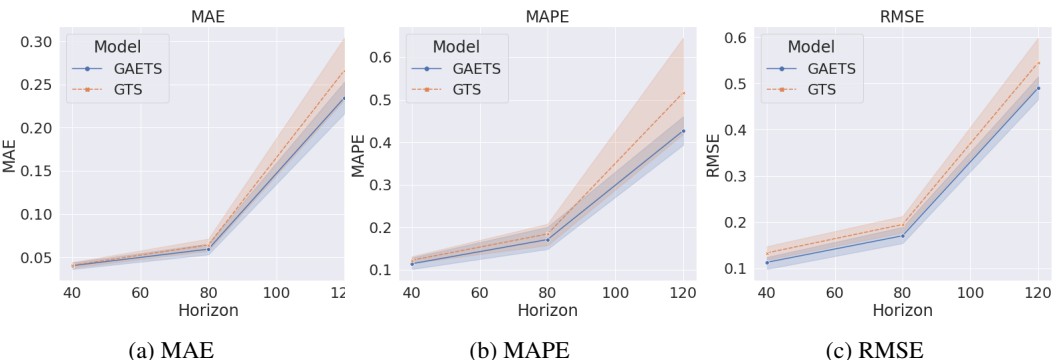

Figure 6: Evaluation metrics comparing the proposed GAETS architecture with GTS

## 5 CONCLUSION

Battery parameter estimation remains to be a challenging and topic of relevance due to the burgeoning number of Electric Vehicles in the past decade. Accurate battery capacity estimation also helps to predict remaining usage time in battery powered consumer electronics. In this paper we introduce a Graph AutoEncoder Time Series (GAETS) architecture which exploits the dependencies between battery parameters (charge capacity, voltage, current etc) to provide precise estimates. We compare our architecture with the existing Graph Time Series architecture to highlight the effectiveness of our approach.

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
