# OpenReview forum: "GAETS: A Graph Autoencoder Time Series Approach Towards Battery Parameter Estimation"
_ICLR.cc/2022/Conference — ICLR 2022 Submitted_

### Official Review · Reviewer_5kWi · 2021-10-21

**Correctness:** 2
**Technical Novelty And Significance:** 2
**Empirical Novelty And Significance:** 2
**Recommendation:** 5
**Confidence:** 3

**Main Review:**

Strengths:
1. The paper is easy to read.
2. In terms of methodology, instead of learning graph structure directly, the method iteratively samples the graph with learned parameters for the encoder-decoder framework. The Gumbel reparameterization technique is used and enables the gradients to flow through A.
3. Experiments outperform the SOTA method and achieve satisfactory results.

Weaknesses:
My major concern of this paper:
1. The technical contribution is limited. Considering that the architecture of RNN+GNN is not novel, the main contribution of this paper may come from the application of graph sampling for learning the relationship between variables.
[KDD20] Wu, Zonghan, et al. "Connecting the dots: Multivariate time series forecasting with graph neural networks."
[AAAI21]Deng, Ailin, and Bryan Hooi. "Graph neural network-based anomaly detection in multivariate time series."

2. The necessity of graph structure learning needs to be proved. The dataset used in this paper has only six dimensions of variables, and the underlying relationships learned by the method are not explicitly shown in this paper.

3. The experimental results are not enough. The proposed method is only compared with GTS, while more baseline methods need to be included. The authors may want to conduct an ablation study to show the effectiveness of each component in their method.
[AAAI2021]Zhou, Haoyi, et al. "Informer: Beyond efficient transformer for long sequence time-series forecasting."

**Summary Of The Paper:**

This paper studies a new problem of battery parameter estimation. The paper proposes a new graph autoencoder time series estimation approach that learns the underlying relationship between variables in battery measurements. Experimental results show that the new method outperforms the SOTA method and achieves satisfactory results.



**Summary Of The Review:**


Other comments:

- The functions of g_1 and g_2 are not explained.

- In 3.1.2, the equation of W_(Q*A)Y is confusing. This function is not mentioned in the rest of the paper.

- In Fig3,4,5, the results of the proposed method compared with the baseline seems not significant.

---

### Official Review · Reviewer_61UC · 2021-10-30

**Correctness:** 2
**Technical Novelty And Significance:** 2
**Empirical Novelty And Significance:** 2
**Recommendation:** 3
**Confidence:** 3

**Main Review:**

Strengths:
- The idea of incorporating causal structure learning with graph-based forecasting is interesting and, to the best of my knowledge, novel.

Weaknesses:
- The motivation of incorporating causal structure learning is unclear. The authors could provide more explanation, and also visualize the learned adjacency matrix to verify if the learned relationships are consistent with domain knowledge.
- In Section 3, $A$ is not guaranteed to be acyclic, different from [1], since the author does not enforce the acyclicity constraint. In this case, how is it related to causality?
- Apart from demonstrating that the proposed method works on the battery dataset, I would encourage the authors to compare it to standard traffic prediction dataset, as also mentioned in Section 2.

Other comments:
- A large part of Figure 2 seems to be similar to Figure 1 in [2] with a different color.
- Also, Equation (1) seems to be similar to Equation (5) in [3] but is not referenced.

References:

[1] DAGs with NO TEARS: Continuous Optimization for Structure Learning, 2018.

[2] Discrete Graph Structure Learning for Forecasting Multiple Time Series, 2021.

[3] A Graph Autoencoder Approach to Causal Structure Learning, 2019.

**Summary Of The Paper:**

The paper proposes to incorporate a graph-autoencoder method from the field of causal structure learning for improving graph-based forecasting, with a focus on battery parameter estimation. In particular, the authors include a regularization term based on graph-autoencoder in the forecasting model to learn the variable dependencies. The authors demonstrate empirically that the proposed method has an improved performance over the baseline considered.

**Summary Of The Review:**

The idea of incorporating causal structure learning with graph-based forecasting is interesting and novel, but the motivation is unclear. Moreover, the adjacency matrix used is not guaranteed to be acyclic, and, in that case, the authors should explain how it is related to causality.

---

### Official Review · Reviewer_LsgQ · 2021-10-30

**Correctness:** 2
**Technical Novelty And Significance:** 2
**Empirical Novelty And Significance:** 2
**Recommendation:** 3
**Confidence:** 4

**Main Review:**

Pros:
1. The problem is meaningful. Lithium-ion batteries are driving the current electrification revolution. Studying the time-series battery estimations is impactful.
2. The method achieves good results on the real-world battery dataset and outperforms the compared SOTA method.

Cons:
1. The proposed method, GAETS, essentially adds a graph autoencoding module on top of the previous GTS method. The autoencoding module acts as a regularizer. But overall this added module is only incrementally novel. The paper lacks enough evidence to support why this autoencoding is necessary.
2. The experiment is thin. There are many existing temporal graph neural networks for time-series forecasting [1]. It is not very convincing when you only compare with 1 other method. Also, as you mentioned in the introduction, there also exist physics-based modeling and phenomenological battery modeling. As an application specific paper, it makes more sense to include those models in comparison.
3. The method looks quite general. Could you also validate its performance on other time-series scenarios?
4. Some descriptions are not very clear. In sec 3.1, when the graph structure $A$ is first introduced, there is no prior descriptions about what is this A. I understand you inherit many setups from GTS paper. But you really need to make sure your paper is self-contained and provide all necessary details. Also, you mentioned $A$ is sampled. How is it sampled should be clarified? Do you sample one $A$ for all timestamps, or sample one $A$ for each timestamp?
5. In your problem, you have the index for different time series sources and the timestamp index for sequence data. You should better clarify the notations and be more consistent. For instance, In sec 3.1.1, you have $X^{(j)}$ and $X^i$. It is confusing whether they denote different things.
6. You only give brief introductions for the recurrent networks. Given its importance in your work, you should elaborate on this section and expand it (even it is from previous works).
7. When you first introduce $A$ is $n\times n$ in sec 3.1, you didn't specify what $n$ is. $R, C, U$ from sec 3.1.2 are not well explained either.


[1] Spectral Temporal Graph Neural Network for Multivariate Time-series Forecasting. Cao, D., Wang, Y., Duan, J., Zhang, C., Zhu, X., Huang, C., Tong, Y., Xu, B., Bai, J., Tong, J. and Zhang, Q. NeurIPS 2020.

**Summary Of The Paper:**

This work proposes a new method based on graph neural network for Lithium-ion batteries parameters estimation. In a typical time-series data, one is given some past timestamps and is asked to predict some future timestamps. In this problem, you have multiple time series of battery parameters. The goal is to predict multiple future states for all the time series data. Since the time series data could be correlated, this work introduces a graph neural network where each time series source is a node on the graph, and the node relations are learnt through training. The graph is used in the recurrent graph convolution and graph autoencoding. The loss function has a future prediction component, and an autoencoding reconstruction component to regularize the learning. The method is validated on a Lithium-ion battery dataset. It outperforms the compared state-of-the-art method.

**Summary Of The Review:**

This paper is an application paper, on Lithium-ion battery parameter estimations. While I generally appreciate the application, I think the paper is thin. Method-wise, it is incrementally novel compared to the prior work GTS. Experiment-wise, given this is an application paper, authors are expected to show more experiments and give more analyses, including more datasets and compared methods. Even the given experiment shows the proposed method is only marginally better than the only compared method. The paper is neither well-organized nor self-contained. The authors should keep improving the paper.

---

### Official Review · Reviewer_L9LB · 2021-11-03

**Correctness:** 4
**Technical Novelty And Significance:** 1
**Empirical Novelty And Significance:** 2
**Recommendation:** 3
**Confidence:** 4

**Main Review:**

The graph auto encoder used in the paper forecasts the state of charge using an absolute error prediction loss. This part is based on an existing algorithm (Shang et al 2021a and 2021b). The neural network architecture is also based on the same paper.

The paper proposes adding a reconstruction loss (given by absolute error) to the prediction loss as a regularizer. This is a small addition to the algorithm. The resulting performance improvement is also marginal as seen by the prediction graphs between the proposed algorithm GAETS and existing algorithm GTS.

Therefore, the paper does not meet the acceptance criteria for a top tier conference such as ICLR.

The paper can also benefit from analysis of the relationships learned by the graph network to inform the domain experts in the area.

**Summary Of The Paper:**

The paper proposes a graph neural network based forecasting algorithm for battery state of charge. Building on prior work, the paper proposes a regularization based on reconstruction loss. Evaluation on a battery dataset shows improvements.

**Summary Of The Review:**

The proposed algorithm is incremental over the existing ones and the corresponding impact on results is marginal.

---

### Decision · Program_Chairs · 2022-01-20

**Decision:**

Reject

**Comment:**

The paper proposes to apply graph neural networks to predict battery state of charge. The main concern is the lack of technical novelty, since the main work is a straightforward application of existing works. The work could be better suited for a more application-oriented venue.